# Quantitative Proteomic Analysis of Zearalenone-Induced Intestinal Damage in Weaned Piglets

**DOI:** 10.3390/toxins14100702

**Published:** 2022-10-13

**Authors:** Lulu Ma, Yanping Jiang, Fuguang Lu, Shujing Wang, Mei Liu, Faxiao Liu, Libo Huang, Yang Li, Ning Jiao, Shuzhen Jiang, Xuejun Yuan, Weiren Yang

**Affiliations:** 1College of Animal Sciences and Veterinary Medicine, Shandong Agricultural University, Tai’an 271018, China; 2Zhongcheng Feed Technology Co., Ltd., Feicheng 271600, China; 3Shandong Yucheng Animal Husbandry Development Center Co., Ltd., Yucheng 251200, China; 4College of Life Sciences, Shandong Agricultural University, Tai’an 271018, China

**Keywords:** zearalenone, weaned piglets, intestine, histology, intestinal permeability, proteomic

## Abstract

Zearalenone (ZEN), also known as the F-2 toxin, is a common contaminant in cereal crops and livestock products. This experiment aimed to reveal the changes in the proteomics of ZEN-induced intestinal damage in weaned piglets by tandem mass spectrometry tags. Sixteen weaned piglets either received a basal diet or a basal diet supplemented with 3.0 mg/kg ZEN in a 32 d study. The results showed that the serum levels of ZEN, α-zearalenol, and β-zearalenol were increased in weaned piglets exposed to ZEN (*p* < 0.05). Zearalenone exposure reduced apparent nutrient digestibility, increased intestinal permeability, and caused intestinal damage in weaned piglets. Meanwhile, a total of 174 differential proteins (DEPs) were identified between control and ZEN groups, with 60 up-regulated DEPs and 114 down-regulated DEPs (FC > 1.20 or <0.83, *p* < 0.05). Gene ontology analysis revealed that DEPs were mainly involved in substance transport and metabolism, gene expression, inflammatory, and oxidative stress. The Kyoto Encyclopedia of Genes and Genomes analysis revealed that DEPs were significantly enriched in 25 signaling pathways (*p* < 0.05), most of which were related to inflammation and amino acid metabolism. Our study provides valuable clues to elucidate the possible mechanism of ZEN-induced intestinal injury.

## 1. Introduction

Zearalenone (ZEN) is one of the most prevalent mycotoxins currently contaminating food and feed synthesized by the fungus *Fusarium* spp. [1]. Zearalenone and its metabolites are classified as xenoestrogens due to their structural similarity to endogenous estrogens [2,3]. Previous studies have shown that ZEN exposure can induce an immune response, oxidative damage, and other pathological changes in different tissues of the body, causing organ damage, decreased productive and reproductive performance, and even acute animal death in severe cases [4,5,6].

The small intestine is the most important site related to nutrient digestion and absorption, and it also serves as the first barrier to defense against pathogen invasion and mycotoxin hazards [7,8]. Previous studies have shown that the intestine is a major target organ of ZEN metabolization, and is highly vulnerable to ZEN-induced damage [9]. Weaning is one of the most stressful periods in the pig’s life that can contribute to intestinal dysfunctions, leading to reduced pig health and growth retardation [10]. Therefore, weaned piglets are more susceptible to ZEN contamination [11,12]. Saenz et al. [13] found that the ingestion of ZEN (679 and 1623 μg/kg) could alter the gut microbiome by increasing intestinal oxidative stress in weaned piglets. Our previous study showed that weaned piglets fed diets containing 1.04 mg/kg ZEN showed decreased activities of disaccharidase enzymes, reduced the intestinal functional mucosal epithelial surface area, and enhanced oxidative stress in the small intestine [14,15]. Zhang et al. [16] also demonstrated that 3.0 mg/kg ZEN exposure damaged cecal morphology and integrity through suppressing the TGF-β1/Smads signaling pathway in weaned piglets. Moreover, a study in vitro also reported that 100 μM ZEN promoted the development of intestinal pathology in castrated male pigs by activating Wnt/β-catenin signaling and inhibiting TGF-β [17]. The overall process of toxin action on cells or tissues is usually accompanied by changes in genomics, metabolomics, and proteomics [18,19]. Solar et al. [19] explored the proteome changes induced by a short, non-cytotoxic exposure to ZEN in the intestine using pig jejunal explants and indicated that ZEN could alter epithelial homeostasis through regulating a cascade of highly inter-connected signaling events essential for the small intestinal crypt–villus cycle. Proteomics is an effective tool for the comprehensive analysis of toxin-infected cells and tissues, which facilitates the elucidation of the potential pathogenic mechanisms of toxins [20]. However, global protein and signaling changes in ZEN-damaged intestines of weaned piglets in vivo trials have not been reported.

Therefore, in this study, the tandem mass tag (TMT) proteomics was utilized to investigate the changes in protein expression and important signaling pathways in the small intestines of weaned piglets treated with 3.0 mg/kg ZEN, uncovering more possible molecular mechanisms of ZEN toxicity in the small intestines of weaned piglets.

## 2. Results

### 2.1. Serum ZEN, α-ZOL, and β-ZOL

Compared with the control, the ZEN treatment had higher serum levels of ZEN, α-zearalenol (α-ZOL), and β-zearalenol (β-ZOL) (*p* < 0.05) (Table 1).

### 2.2. Apparent Nutrient Digestibility

As shown in Table 2, compared to the control, the apparent digestibility of ether extract (EE) and crude protein (CP) was significantly lower in the ZEN piglets (*p* < 0.05), while no significant differences were observed in the apparent digestibility of dry matter (DM) and organic matter (OM) between the two groups (*p* > 0.05).

### 2.3. Histopathological Examination

Histopathological examination was used to observe the damage caused by ZEN in jejunum tissue. As shown in Figure 1A, compared to the control, the jejunum villi were significantly damaged and disorganized, and the lamina propria of the epithelium was detached in the ZEN. Piglets in the ZEN group had a significantly decreased jejunal villus height and the ratio of villus height to crypt depth (*p* < 0.05) and had significantly increased crypt depth compared to the control (*p* < 0.05) (Figure 1B).

### 2.4. Intestinal Permeability of Weaned Piglets

The effect of ZEN on the intestinal permeability of weaned piglets is shown in Table 3. Piglets in the ZEN group had significantly higher serum endotoxin (ET), diamine oxidase (DAO), and D-lactate contents than piglets in the control group (*p* < 0.05).

### 2.5. Quantitative Mapping of Jejunum Proteome in the Control and ZEN Treatment

To identify proteomic changes of ZEN-induced damages in the jejunum of piglets, a powerful TMT-labeled quantitative proteomics analysis technique was applied in this experiment. A total of 306,099 bands, 39,827 peptide numbers, and 7145 proteins (Appendix A) were identified and obtained. A total of 174 differentiable expressed proteins (DEPs, fold change (FC) > 1.20 or < 0.83, *p* < 0.05) were identified in the ZEN compared with the control (Appendix A), including 60 up-regulated DEPs and 114 down-regulated DEPs (Figure 2A). Notably, we identified 22 uncharacterized or unassigned proteins among the 174 DEPs, because the current porcine genome database was not fully annotated compared to the human genome database. Therefore, the functional analysis of these proteins warrants further investigation.

As shown in Figure 2C,D, six proteins were randomly selected from the DEPs to validate the results of the TMT proteomic analysis, including serum amyloid P-component (APCS), 15-oxoprostaglandin 13-reductase (PTGR1), glutamate carboxypeptidase 2 (FOLH1), bifunctional glutamate/proline-tRNA ligase (EPRS), eukaryotic elongation factor 2 kinase (EEF2K), and S100 calcium-binding protein A16 (S100A16). The results of the Western blot were in high agreement with the data of the TMT analysis, indicating the high reliability of the results of this proteomics analysis (Figure 2B).

### 2.6. Top Ten Up-Regulated or Down-Regulated DEPs

The top ten up-regulated proteins in the ZEN group are listed in Table 4. Among the top ten up-regulated proteins, the SERPIN domain-containing protein and uncharacterized protein exert enzyme inhibitory activity; the glutathione S-transferase kappa 1 isoform a (fragment), 15-oxoprostaglandin 13-reductase, and the threonyl-tRNA synthetase exert enzymatic activity; the beta-parvin isoform X3 binds to actin; and the apolipoprotein C-II exerts enzyme activator activity. The main functions of the Ig-like domain-containing protein, uncharacterized protein, and nuclear autoantigenic sperm protein have not been reported.

The top ten down-regulated proteins in the ZEN group are listed in Table 5. Of the top ten down-regulated proteins, the glutathione S-transferase kappa 1 isoform a (fragment), enoyl-CoA hydratase, glutamyl-tRNA synthetase, and threonyl-tRNA synthetase exert enzymatic activity; the EF-hand domain-containing protein binds to calcium ions; the lysozyme C-2 plays an immune-enhancing role; the translation initiation factor eIF-2B subunit delta isoform 2 exerts translation initiation factor activity; the eukaryotic elongation factor 2 kinase binds to calmodulin; the HIT domain-containing protein plays a catalytic activity. In addition, there is one down-regulated protein whose function has not been described.

### 2.7. Gene Ontology (GO) Functional Annotation and Enrichment Analysis of DEPs

To characterize the DEPs, we performed GO functional annotation statistics on the DEPs, which were divided into three major categories at the functional level: biological process (BP), cellular component (CC), and molecular function (MF) [21].

Within the BP category, the DEPs were predicted to be linked with 16 biological processes, such as cellular process, biological regulation, and metabolic process (Figure 3, Appendix A). Among them, proteins associated with the regulation of the modulation of the molecular function in other organisms, the modulation of the molecular function in other organisms involved in symbiotic interaction, modulation by a host of symbiont molecular function, response to thyroid hormone, and cellular response to thyroid hormone stimulus were the top five significantly enriched in the BP category (Figure 4, Appendix A).

Within the CC category, the DEPs were predicted to be primarily distributed within two different cellular components, including the cellular anatomical entity and protein-containing complex (Figure 3, Appendix A). Especially, proteins associated with the regulation of nuclear pericentric heterochromatin, chromocenter, pericentric heterochromatin, histone deacetylase complex, and nuclear speck were the top five significantly enriched in the CC category (Figure 4, Appendix A).

Additionally, within the MF category, the DEPs were predicted to be linked with ten molecular functions, for instance, binding, catalytic activity, and the molecular function regulator (Figure 3, Appendix A). Particularly, proteins associated with the regulation of proteoglycan binding, supercoiled DNA binding, threonine-tRNA ligase activity, glutathione peroxidase activity, and SH3 domain binding were the top five significantly enriched in the MF category (Figure 4, Appendix A).

### 2.8. Kyoto Encyclopedia of Gene and Genomes (KEGG) Enrichment Analysis of DEPs

KEGG enrichment analysis was performed on 174 DEPs, and a total of 25 significantly different pathways were enriched for these proteins. Those pathways related to inflammation and amino acid metabolism were more enriched. The major pathways related to inflammation included the Rap1 signaling pathway, B cell receptor signaling pathway, and Fc gamma R-mediated phagocytosis. These three pathways contain two identical DEPs, including the Ig-like domain-containing protein (LOC100523213) and protein kinase domain-containing protein (PBK). The pathways related to amino acid metabolism mainly included lysine degradation, beta-alanine metabolism, and tryptophan metabolism (Figure 5, Appendix A). The aldehyde dehydrogenase (NAD^+^) (ALDH-NAD^+^) of DEPs was involved in the three pathways.

## 3. Discussion

Zearalenone can be converted to α-ZOL and β-ZOL after being absorbed by the intestine. Once the rate of ZEN deposition exceeded that of metabolism, ZEN and its metabolites might be accumulated in the body [22,23,24]. However, a previous study showed that the positive detection rate of ZEN could reach 69.15%, and the highest value of ZEN in compound feed samples was 4.33 mg/kg [6]. In our study, significantly increased serum ZEN, α-ZOL, and β-ZOL levels of weaned piglets were observed in the piglets fed the diet containing 3.0 mg/kg ZEN, which was in line with our previous study [25]. Zhang et al. [24] found that 970 μg/kg ZEN could increase the levels of ZEN, α-ZOL, and β-ZOL of serum in gilts. Moreover, it was also reported that oral 1.0 mg/kg BW ZEN increased serum levels of ZEN, α-ZOL, and β-ZOL in juvenile female pigs [26]. Long-term ZEN deposition can damage intestinal development and function, negatively influencing the digestive system of animals [27]. In the present study, the apparent digestibility of CP and EE in ZEN piglets was significantly reduced. Consistently, our previous study in piglets showed that the apparent digestibility of CP, gross energy (GE), the metabolic rate of GE, and net protein utilization were decreased by ZEN (more than 1.0 mg/kg) treatment [28]. Wang et al. [29] also reported that the apparent digestibility of DM and nitrogen was decreased with increasing concentrations of ZEN (0.2 to 0.8 mg/kg) in the weaned piglets. 

The small intestine is the principal organ in charge of nutrient absorption, and intestinal morphology and structural integrity were fundamental to maintaining normal function [9]. In this study, ZEN exposure disrupted the jejunal villus and glands and decreased the villus height and villus height/crypt depth ratio, showing intestinal barrier damage and nutrient absorption area reduction. The villi height, crypt depth, and the villi height/crypt depth ratio were considered important indicators to evaluate the ability of the animal to digest and absorb nutrients [30]. The shortening or loss of the intestinal villus usually led to a reduction in nutrient absorption area, resulting in malnutrition, diarrhea, and decreased disease resistance [31]. Liu et al. [32] also found that ZEN (0.3 to 146 mg/kg) damaged the jejunal villus of pregnant dams in a dose-dependent manner, leading to a reduction in functional mucosal surface area. Moreover, a previous study in rats showed that 1.0 and 5.0 mg/kg of ZEN caused intestinal villus and gland injury with the separation of the submucosa and lamina propria [9]. In addition, in the present study, we found that ZEN exposure increased the serum concentrations of ET, DAO, and D-lactate in weaned piglets. A study in rats also proved that ZEN (1.0 and 5.0 mg/kg) exposure significantly elevated serum DAO and D-lactate concentrations [9]. Endotoxin was a component of the cell wall of Gram-negative bacteria [33], and D-lactate was a specific metabolite of intestinal bacteria [34]. The DAO was an intracellular enzyme present in Mammalian intestinal mucosal cells [35]. The ET, DAO, and D-lactate would be released into the circulation system when the intestinal barrier was compromised [36]. Therefore, Blood ET, DAO, and D-lactate levels were considered key markers to assess intestinal permeability [36]. Above all, our results suggested that ZEN induced the decrease in the apparent digestibility of nutrients in weaned piglets partly through damaging the small intestinal morphology and barrier integrity.

To characterize the underlying mechanisms of ZEN-induced intestinal damage in weaned piglets, the TMT proteomics approach was employed in this study. We further confirmed the results of TMT proteomics using Western blot and found the related proteins significantly altered with up-regulated APCS, PTGR1, and FOLH1 expressions and down-regulated EPRS, EEF2K, and S100A16 expressions in ZEN piglets. Gene ontology and KEGG enrichment analysis provide important references and clues for understanding the functions of DEPs [37]. In the present study, GO terms and signaling pathways related to inflammation and substance metabolism were more enriched with ZEN treatment. Consistently, Gajęcka et al. [38] indicated that ZEN stimulated energy and protein metabolic processes in pre-pubertal gilts. Besides, a previous study in mice showed that ZEN increased intestinal inflammatory factors [39]. Therefore, inflammatory response and metabolic disorders induced might be contributed to the intestinal injury induced by ZEN in this study.

In our study, almost half of the signaling pathways were related to inflammation. The inflammatory pathways of the Rap1 signaling pathway, B cell receptor signaling pathway, and Fc gamma R-mediated phagocytosis were activated by ZEN. The Rap1 signaling pathway was activated to increase cell adhesion, polarization, and chemotaxis, and could regulate the inflammatory process by activating monocytes [40]. Cai et al. [41] demonstrated that the Rap1 could induce cytokine production in pro-inflammatory macrophages through NF-κB signaling in human atherosclerotic lesions. The function and maturation of B cells receptor (BCR) were closely connected with the proliferation and differentiation of B cells [42]. BCR-mediated antigen recognition was thought to regulate B cell differentiation, and the activation and amplification of BCR signaling benefited to promote B cell survival and growth [43,44]. Fcγ receptors (FcγRs), receptors for IgG, classically regulated the course of the immune response [45] and B cell activation signals delivered by the BCR [46]. Interestingly, two identical DEPs, LOC100523213 and PBK, were both involved in the three pathways. In this study, ZEN exposure up-regulated the LOC100523213 expression and down-regulated the PBK expression in the intestine. These results further demonstrated that ZEN could induce intestinal damage by activating the inflammatory response in the intestine of weaned piglets. The LOC100523213 belongs to the immunoglobulin superfamily [47]. Inflammation stimulation or infection could result in the activation of B cells and increased immunoglobulins, which improves the ability of the body to remove pathogens that induce humoral immune responses [48,49]. A recent study reported that the LOC100523213 expression was elevated in the serum of meningitis piglets [50]. The PBK is a novel serine/threonine kinase that is highly expressed in proliferating cells and tissues [51]. A recent study demonstrated a negative correlation between PBK expression and immune suppressor cells, including regulatory T cells and M2 macrophages [52]. Moreover, Zhu et al. [53] found that PBK expression was suppressed after the development of inflammation in thin endometrium. Therefore, ZEN-induced inflammatory response might be one reason for intestinal injury in weaned piglets.

Other highly enriched signaling pathways were associated with amino acid metabolism. In this study, the amino acid metabolic pathways of lysine, beta-alanine, and tryptophan were activated under ZEN treatment. The intestine is the main site of amino acid absorption and metabolism [54]. The activation of amino acid metabolic pathways further demonstrated that ZEN could regulate amino acid absorption and metabolism in weaned piglets. Lysine and tryptophan are not only essential amino acids but also serve as the first and second limiting amino acids for pigs [55,56]. Lysine and tryptophan deficiencies in piglet diets could cause reduced feed intake and feed utilization and suppress immune function, leading to increased morbidity and mortality [57,58,59,60]. A previous study also indicated that the ileal apparent digestibility of tryptophan was significantly reduced in piglets feeding the basal diet supplemented with 10 mg/kg of ZEN compared with those fed the basal diet [61]. Beta-alanine was the precursor for the synthesis of functional compounds such as coenzyme A and pantothenic acid [62]. As a potentially functional amino acid, beta-alanine plays an important role in maintaining the normal growth of organisms [63,64]. In addition, the ALDH-NAD^+^ from the three pathways mentioned above was suppressed in the ZEN-treated piglets. The ALDH-NAD^+^ was the one of ALDH [65], and its deficiency enhanced the ethanol-induced disruption of intestinal epithelial tight junctions and barrier dysfunction [66]. Therefore, we hypothesized that ZEN could cause damage to the intestine by regulating amino acid metabolic processes via inhibiting ALDH expression in weaned piglets. However, precise functions of those pathways and DEPs in ZEN-induced intestinal damage remain to be elucidated by cellular experiments.

## 4. Conclusions

In summary, our study showed that 3.0 mg/kg ZEN resulted in decreased nutrient digestibility and destroyed intestinal integrity. Meanwhile, ZEN could damage the intestine of weaned piglets through changing the process of substance metabolism and triggering an inflammatory response. Additionally, the DEPs, acting as intermediates or key enzymes in a variety of potential signaling pathways, were mainly involved in pathways related to inflammation and amino acid metabolism in this study. Although further studies will be required to elucidate the functions of the DEPs, our study provides valuable clues to elucidate the possible mechanism of ZEN-induced intestine intestinal injury and lays the groundwork for future research on ZEN detoxification in animals. 

## 5. Materials and Methods

### 5.1. Animals, Treatments, and Feeding Management

A total of 16 healthy Duroc × Landrace × Yorkshire weaned piglets (35 d of age) with an average body weight (BW) of 12.45 ± 0.19 kg were randomly allotted to two treatments with eight replicates per treatment and one piglet per replicate. Weaned piglets either received a basal diet (control group) or a basal diet supplemented with 3.0 mg/kg ZEN (ZEN group). The dose of ZEN was selected according to the results of our previous study that 3.0 mg/kg of ZEN in piglet diets induced the obstruction of intestinal self-repair [16]. The basal diet was prepared according to the National Research Council (NRC, 2012, Washington, DC, USA), and the ration formulation and its contents of various major nutrients were shown in Appendix A. The weaned piglets used in this trial were housed in cages with an area of 0.48 m^2^ for a 32 d experiment period after a 7 d adjustment. The room temperature was controlled at 30 °C in the first week of the experiment and then maintained between 26 °C and 28 °C until the end of the experiment. The relative humidity was kept at about 65%.

The mycotoxins of the diets were carried out by Qingdao Entry-Exit Inspection and Quarantine Bureau (Qingdao, China) as previously described [67]. The levels of ZEN and aflatoxin were quantified by liquid chromatography in conjunction with fluorescence detection, affinity column chromatography, and external standard method, and the contents of fumonisin and deoxynivalenol were quantified by high-performance liquid chromatography–tandem mass with fluorescence detection, affinity column chromatography, and external standard method [14,16]. The minimum detection limits of ZEN, fumonisins, deoxynivalenol, and aflatoxin were 0.01 mg/kg, 0.25 mg/kg, 0.1 mg/kg, and 1.0 μg/kg, respectively. The analyzed ZEN levels in the basal diet and ZEN diet were <0.01 and 3.12 ± 0.13 mg/kg, respectively, and the other toxins were not detected.

### 5.2. Sample Collection

Piglets were fasted for 12 h before sampling on the last day of the trial. Approximately 5 mL of blood was collected from the jugular veins into vacuum procoagulation tubes and placed at room temperature for 20 min. The serum was obtained after being centrifuged at 3000× *g* for 15 min in a low-temperature centrifuge and stored at –20 °C immediately. The serum was used to determine the toxin, ET, DAO, and D-lactate content. After blood samples collection, the piglets were injected intramuscularly with 0.1 mg/kg BW Zoletile 50 Vet (Virbac, Brittany, Carros, France) and immediately slaughtered for jejunum sampling. One part of the jejunum samples was fixed in 4% paraformaldehyde solution for 24 h to observe the morphological changes of the intestine, while the other part of the jejunum samples was stored in a −80 °C refrigerator for antioxidant capacity, protein expression, and proteomic analysis.

### 5.3. Serum ZEN, α-ZOL, and β-ZOL

Serum ZEN, α-ZOL, and β-ZOL were determined by the Institute of Quality Standards and Detection Technology of the Chinese Academy of Agricultural Sciences (Beijing, China). The specific operation method was referenced by Wan et al. [25].

### 5.4. Apparent Nutrient Digestibility

Feces excreted by the piglets were collected daily for four consecutive days starting on day 20 of the experiment. The feces were weighed and mixed, and representative samples were stored in a −20 °C refrigerator after the nitrogen was fixed with 10% sulfuric acid to determine CP (Kjeldahl method). The rest of the feces were baked to constant weight in a 65 °C thermostat, followed by being crushed and stored in sealed containers for the determination of DM (105 °C drying method), crude ash (CA, 550 °C scorching method), OM (OM = DM − CA), and EE (Soxhlet extraction method). The apparent nutrient digestibility of DM, OM, CP, and EE was calculated in a previous study [68].

### 5.5. Histopathological Examination of Jejunum

After being fixed in 4% paraformaldehyde solution for 24 h, the jejunum samples were trimmed and placed in an embedding box. Then, they were rinsed in running water for 48 h, followed by alcohol gradient dehydration, xylene transparency, and paraffin embedding. Finally, serial sections of 5 μm thickness were created and dewaxed in a gradient solution of xylene and ethanol after being dried at 37 °C, followed by staining with hematoxylin and eosin. The jejunum structures were observed using an Olympus BX41 microscope equipped with a DP25 digital camera (Olympus, Tokyo, Japan). The intestinal villus height and crypt depth of weaned piglets were carried out using Motic images 2000 software (version 1.3, Motic Incorporation, Ltd., Hong Kong, China). Eight sections were chosen from each sample, and ten well-extended, straight, and intact villi were selected from each section for measurement. The villi height and the crypt depth around them were measured and averaged to calculate the ratio of villi height to crypt depth. The detail determination methods of the villi height and the crypt depth were referred to in a previous study [35]. In short, the villi height was measured from the villi tip to the villi base, and the crypt depth was measured from the intervillous valley to the basement membrane.

### 5.6. Intestinal Permeability of Weaned Piglets

The levels of ET and DAO in serum were determined using commercial ELISA assay kits purchased from the Jiangsu Meimian Industrial Co., Ltd., (Yancheng, China). The ELISA experiment procedures were derived from Chen et al. [69]. Additionally, the D-lactate content of the serum was measured by a corresponding ELISA assay kit (Nanjing Jiancheng Bioengineering Institute, Nanjing, China). The ELISA experiment procedures were referenced by Gao et al. [70].

### 5.7. Sample Preparation and TMT Labeling

To determine the expression of the entire proteome, eight jejunum samples were selected for TMT-based quantitative proteomic analysis in Majorbio Bio-pharm Technology Co., Ltd., (Shanghai, China). The screenshot of the workflow of Proteome Discoverer was shown in Appendix A. Four jejunum samples were randomly selected from each group. The samples were removed from the refrigerator at –80 °C and transferred to centrifuge tubes, followed by being shaken and mixed with protein lysis solution (8 M urea + 1% SDS with protease inhibitor). The tubes were placed in a high-throughput tissue grinder (Tissuelyser-24, Shanghai, China) and lysed on ice for 30 min. Afterward, the tubes were centrifuged (4 °C, 16,000× *g*, 30 min) in a low-temperature centrifuge to separate the supernatants, and the desired protein samples were obtained. The extracted proteins were quantified using the Thermo Scientific Pierce BCA kit (Waltham, MA, USA), and the protein was separated by SDS-polyacrylamide gel electrophoresis (SDS-PAGE).

The 100 μg of protein sample was placed in a centrifuge tube and the volume was replenished with lysate to 90 μL. Tris (2-carboxyethyl) phosphine (TCEP) at a final concentration of 10 mM was added to each tube and incubated for 1 h at 37 °C. Iodoacetamide (IAM) at a final concentration of 40 mM was added to each tube, and the reaction was carried out for 40 min at ambient temperature and protected from light. Then, pre-chilled acetone (acetone: sample volume ratio = 6:1) was added to each tube and precipitated at –20 °C for 4 h. After centrifugation at 10,000× *g* for 20 min at 4 °C, the supernatant was removed, and 100 μL of 100 mM ammonium bicarbonate (TEAB) was added to the pellet. Trypsin (2 µg/µL) was added to each tube at a ratio of trypsin: protein = 1:50 and digested overnight at 37 °C, and then terminated by the addition of 1% trifluoroacetic acid. Acetonitrile was added to TMT reagent (art. NO. 90111, Thermo Fisher, Waltham, MA, USA), which was added to the previously prepared peptide after centrifugation and reacted at room temperature for 2 h. Hydroxylamine was added to each tube and left at room temperature for 30 min. The control group samples were labeled as TMT10-127N, TMT10-127C, TMT10-128N, and TMT10-128C, and the ZEN group samples were labeled as TMT10-129N, TMT10-129C, TMT10-130N, and TMT10-130C, respectively.

### 5.8. High pH Reversed-Phase Liquid Chromatography (HPLC) Fractionation and Liquid Chromatography–Tandem Mass Spectrometry (LC−MS/MS) Analysis

Protein samples were resolubilized with 2% acetonitrile (pH = 10), and high pH liquid phase (Thermo Scientific Vanquish Flex Binary UHPLC system, Waltham, MA, USA) separation was performed using a reversed-phase C18 column (ACQUITY UPLC BEH C18 Column 1.7 µm, 2.1 mm X 150 mm, Waters, Waltham, MA, USA): the A phase was 2% acetonitrile (pH = 10); the B phase was 80% acetonitrile (pH = 10); the UV detection wavelength was 214 nm; the flow rate was 200 μL/min; the gradient: 0–5%B (0–2 min), 5–10%B (2–18 min), 10–30%B (18–35.5 min), 30–36%B (35.5–38 min), 36–42%B (38–39 min), 42–100%B (39–44 min), and 0%B (44–48 min).

After the high pH liquid phase separation, 20 fractions were collected for each sample, combined into 10 fractions, concentrated by vacuum centrifugation, and dissolved in mass spectrometry loading buffer (2% acetonitrile, 0.1% formic acid) for secondary analysis. Each fraction (2 μg protein) was analyzed by chromatography (EASY-nLC 1200, Thermo, Waltham, MA, USA) and mass spectrometry (Q_Exactive HF-X, Thermo, Waltham, MA, USA) using a reversed-phase column (C18 column, 75 μm × 25 cm, Thermo, Waltham, MA, USA): the chromatographic analysis time was 120 min; the A phase was 2% acetonitrile 0.1% formic acid; the B phase was 80% acetonitrile 0.1% formic acid; the flow rate was 300 nL/min; gradient: 5–23%B (0–65 min), 23–29%B (65–79 min), 29–38%B (79–88 min), 38–48%B (88–90 min), 48–100%B (90–92 min), and 100–0%B (92–120 min). The MS scan range (*m*/*z*) was 350 to 1500, and the acquisition mode was the data-dependent acquisition mode. The 20 strongest signals in the parent ions were selected for secondary fragmentation. The primary mass spectrometry resolution was 120,000, the automatic gain control (AGC) target was 3 × 10^6^, the maximum injection time was 50 ms, and the fragmentation mode had higher-energy collision dissociation with 30% normalized collision energy. The secondary resolution was 45,000, the AGC target was 2 × 10^5^, the maximum injection time was 120 ms, the fixed first mass was 110 m/z; the minimum AGC target was 1 × 10^4^, the intensity threshold was 8.3 × 10^4^, and the dynamic exclusion time was 20 s.

### 5.9. Sequence Database Searching and Protein Identification

The RAW files from the mass spectrometry downstream were searched using the Proteome Discoverer^TM^ Software 2.4 (Thermo, Waltham, MA, USA). The RAW files were submitted to the Proteome Discoverer server; the selected species was *Sus scrofa*, the database was uniprot-taxonomy-9823.unique.fasta, and then the database search was performed. The false discovery rate (FDR) of peptide identification during the library search was set to FDR ≤ 0.01, and the protein contained at least one specific peptide. The relevant parameters were shown in Table 6.

### 5.10. Bioinformatics Analysis

GO (https://www.biobam.com/blast2go/ (accessed on 28 July 2021); http://geneontology.org/ (accessed on 28 July 2021)) was selected for functional clustering analysis of all differential proteins. The metabolic pathways involved in differential proteins were analyzed using the KEGG (http://www.genome.jp/kegg/ (accessed on 28 July 2021)) pathway database, which was used to analyze the metabolic pathways involved in the DEPs. Fisher’s exact test was used to identify the GO terms or KEGG pathways that were significantly enriched in significantly differential proteins compared to the proteomic background. *p* < 0.05 can be used as the threshold for significantly elevated GO or KEGG.

### 5.11. Western Blot

The proteins extracted from proteomics were electrophoresed by SDS-PAGE and transferred onto 0.45 μm polyvinylidene fluoride blotting (PVDF) membranes (Millipore, Billerica, MA, USA) by the wet transfer method. After blocking with protein-free rapid blocking buffer (Shanghai Epizyme Biomedical Technology Co., Ltd., Shanghai, China) for one hour at room temperature, the diluted primary antibody anti-APCS (1:500, PA001898ESR1HU, CUSABIO, Wuhan, China), anti-PTGR1 (1:500, PA018983LA01HU, CUSABIO), anti-FOLH1 (1:1000, RA162924A0HU, CUSABIO), anti-EPRS (1:500, PA007747LA01HU, CUSABIO), anti-EEF2K (1:500, PA007435LA01HU, CUSABIO, Wuhan, China), anti-S100A16 (1:500, PA853418LA01HU, CUSABIO), and anti-GAPDH (1:5000, HX1832, Huaxingbio, Beijing, China) were added and incubated overnight at 4 °C. The PVDF membrane was washed three times with TBST (Beyotime, Shanghai, China), and the diluted HRP-labeled Goat Anti-Rabbit IgG (1:2000, A0208, Beyotime, Shanghai, China) was added and incubated for two hours at room temperature. Blots were detected using the ECL chemiluminescence method, which was quantified with Fusion 16.0.9.0 software (Vilber Lourmat, Paris, France).

### 5.12. Statistical Analysis

To detect the difference between the control and the ZEN group, the data were statistically analyzed using an independent samples *t*-test with SAS 9.4 statistical software (SAS Institute Inc., Cary, NC, USA). The data are presented as means ± standard error. The difference was considered significant when *p* < 0.05, and the protein features were considered to be significantly changed between different jejunum samples using a statistical *p* < 0.05 and FC > 1.20 or <0.83.

## Figures and Tables

**Figure 1 toxins-14-00702-f001:**
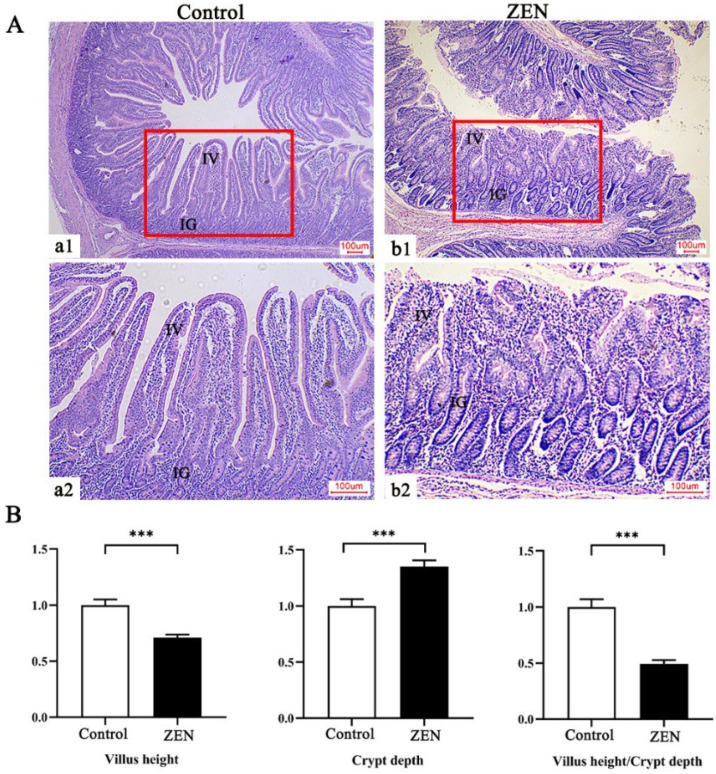
Effects of zearalenone (ZEN) on jejunum morphology and related indexes of weaned piglets (*n* = 8). (**A**) Light micrograph of jejunum section in weaned piglet. (**B**) Morphometric analysis of the jejunum in weaned piglets. The red frames represent the parts of (a1,b1) enlarged to (a2,b2) respectively. The (a1,b1) and (a2,b1) represent the jejunal structures at different magnifications (×40 and ×100), respectively. IV denotes small intestinal villi; IG denotes small intestinal glands. Scale bars were 100 µm for a–b. *** means *p* < 0.001.

**Figure 2 toxins-14-00702-f002:**
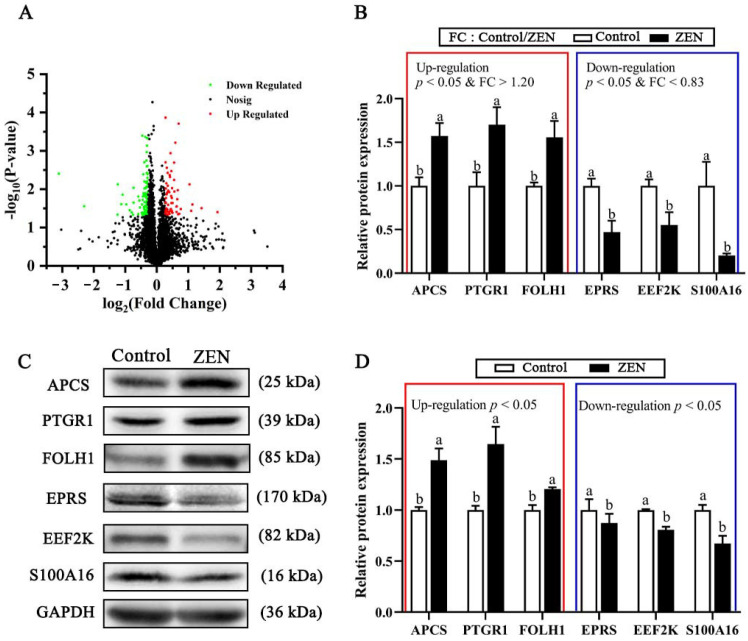
Protein overview of proteomics (*n* = 4) and Western blot (*n* = 3) data. (**A**) Green dots represent the down-regulated proteins meeting the criterion of *p* < 0.05 and fold change (FC, ZEN/Control) < 0.83 compared with the control. Red dots represent the up-regulated proteins meeting the criterion of *p* < 0.05 and FC > 1.20. Black dots indicate that there are no significant changes in protein expression between the Control and ZEN groups. (**B**) Relative expressions of serum amyloid P-component (APCS), 15-oxoprostaglandin 13-reductase (PTGR1), glutamate carboxypeptidase 2 (FOLH1), bifunctional glutamate/proline-tRNA ligase (EPRS), and eukaryotic elongation factor 2 kinase (EEF2K), and S100 calcium-binding protein A16 (S100A16) analyzed by TMT proteomics. The red and blue boxes represent up-regulated and down-regulated differentially expressed proteins, respectively. The same in (**D**). (**C**,**D**) The relative abundance of proteins between the control and ZEN groups analyzed by Western blot. ^a, b^ Means differ significantly (*p* < 0.05).

**Figure 3 toxins-14-00702-f003:**
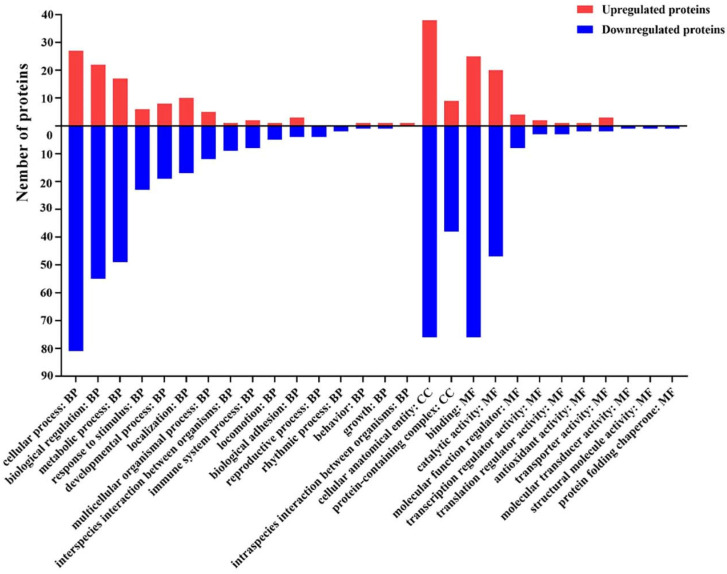
GO functional annotation of 174 significantly differentially expressed proteins (DEPs) in the control and ZEN (*n* = 4). GO annotations for the up-regulated and down-regulated DEPs. The horizontal text indicates the name and classification of the GO terms. Take 0 as the dividing line, above the dividing line are the up-regulated DEPs, below the dividing line are the down-regulated DEPs, and the ordinate indicates the number of up-regulated and down-regulated DEPs (Fisher’s exact test).

**Figure 4 toxins-14-00702-f004:**
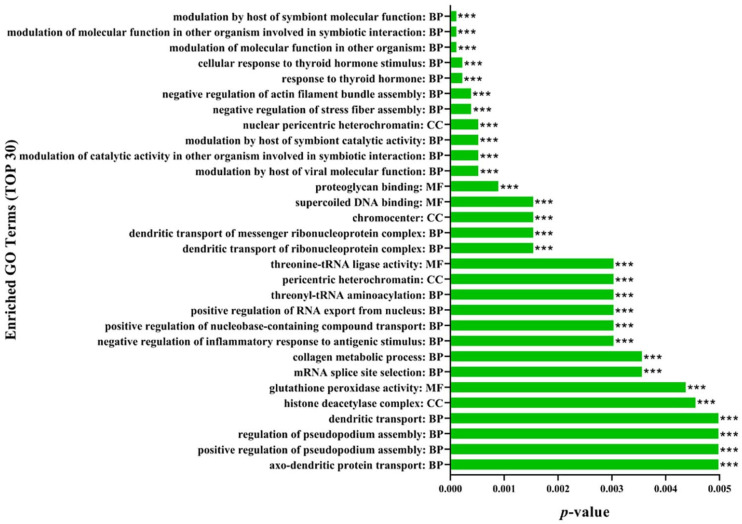
GO enrichment analysis of 174 significantly differentially expressed proteins in the control and ZEN (*n* = 4). The graph shows the enrichment level in the top 30 GO terms. The name and classification of each GO term are indicated in the decimal point (Fisher’s exact test; *** means *p* < 0.001).

**Figure 5 toxins-14-00702-f005:**
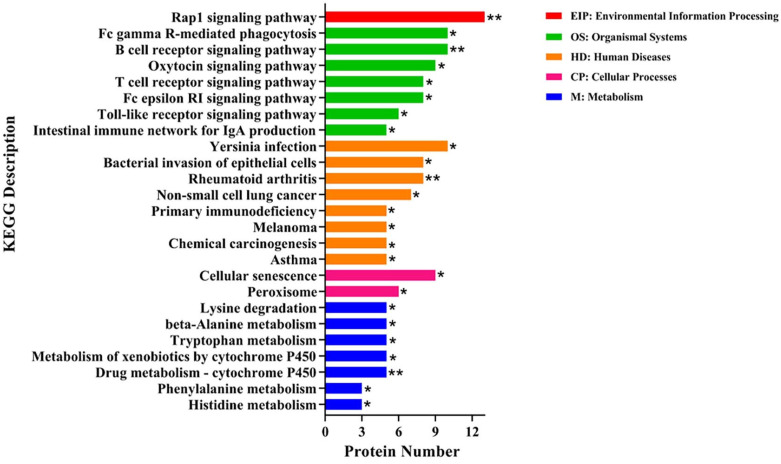
KEGG enrichment analysis of 174 differentially expressed proteins in the control and ZEN (*n* = 4). The horizontal text indicates the name and classification of the KEGG pathway. The KEGG pathway is classified into six categories, including organismal systems (OS), metabolism (M), human diseases (HD), environmental information processing (EIP), and cellular processes (CP), Fisher’s exact test; * means *p* < 0.05, ** means *p* < 0.01.

**Table 1 toxins-14-00702-t001:** Effects of zearalenone (ZEN) on serum toxins of weaned piglets ^1^.

Items	Control	ZEN	*p*-Value
ZEN, ng/mL	0.00 ± 0.00	0.20 ± 0.01	<0.001
α-Zearalenol, ng/mL	0.00 ± 0.00	1.18 ± 0.04	<0.001
β-Zearalenol, ng/mL	0.00 ± 0.00	1.17 ± 0.05	<0.001

^1^ Treatments were basal diet supplemented with ZEN at the level of 0 and 3 mg/kg and with analyzed ZEN concentrations of 0 and 3.12 ± 0.13 mg/kg, respectively. Data are mean value ± standard error (*n* = 8). *p* < 0.05 means differ significantly.

**Table 2 toxins-14-00702-t002:** Effects of zearalenone (ZEN) on the apparent nutrient digestibility of weaned piglets ^1^.

Items	Control	ZEN	*p*-Value
Dry matter, %	89.16 ± 0.57	87.65 ± 1.62	0.398
Organics matter, %	89.55 ± 0.59	86.54 ± 1.98	0.167
Ether extract, %	88.85 ± 1.39	84.24 ± 0.74	0.011
Crude protein, %	86.04 ± 0.72	83.10 ± 0.83	0.018

^1^ Treatments were basal diet supplemented with ZEN at the level of 0 and 3 mg/kg and with analyzed ZEN concentrations of 0 and 3.12 ± 0.13 mg/kg, respectively. Data are mean value ± standard error (*n* = 8). *p* < 0.05 means differ significantly.

**Table 3 toxins-14-00702-t003:** Effects of zearalenone (ZEN) on the intestinal permeability of weaned piglets ^1^.

Items	Control	ZEN	*p*-Value
Endotoxin, U/mL	0.13 ± 0.00	0.67 ± 0.02	<0.001
Diamine oxidase, U/mL	20.28 ± 0.59	24.84 ± 0.72	<0.001
D-lactate, μg/mL	4.98 ± 0.06	5.65 ± 0.09	<0.001

^1^ Treatments were basal diet supplemented with ZEN at the level of 0 and 3 mg/kg and with analyzed ZEN concentrations of 0 and 3.12 ± 0.13 mg/kg, respectively. Data are mean value ± standard error (*n* = 8). *p* < 0.05 means differ significantly.

**Table 4 toxins-14-00702-t004:** Top 10 up-regulated proteins in zearalenone (ZEN) compared with Control ^1^.

Accession	Protein Name	FC	*p*-Value	Main Function
F1SCC9	SERPIN domain-containing protein (LOC106504545)	3.8176	0.0398	serine-type endopeptidase inhibitor activity
A0A480Q1D2	Beta-parvin isoform X3	2.6862	0.0315	actin binding
A0A4 X 1SK13	Ig-like domain-containing protein	2.1993	0.0250	N/A
A0A480THA1	Glutathione S-transferase kappa 1 isoform a (Fragment)	2.1353	0.0368	glutathione peroxidase activity
A0A4X1UQN8	Uncharacterized protein (FRA10AC1)	2.0615	0.0075	N/A
A0A287B7R3	15-oxoprostaglandin 13-reductase (PTGR1)	1.7014	0.0330	prostaglandin reductase activity
A0A480F5A7	Threonyl-tRNA synthetase	1.6397	0.0441	threonine-tRNA ligase activity
A0A4X1UWY7	Uncharacterized protein	1.6200	0.0002	serine-type endopeptidase inhibitor activity
A0A481C6Y8	Nuclear autoantigenic sperm protein	1.6020	0.0452	N/A
A0A4X1W295	Apolipoprotein C-II (APOC2)	1.5911	0.0107	enzyme activator activity

^1^ FC means fold change in ZEN/Control (*n* = 4). N/A indicates that the main function of the protein is not yet clear.

**Table 5 toxins-14-00702-t005:** Top 10 down-regulated proteins in zearalenone (ZEN) compared with Control ^1^.

Accession	Protein Name	FC	*p*-Value	Main Function
A0A480L9T0	Glutathione S-transferase kappa 1 isoform a (Fragment)	0.1161	0.0040	glutathione peroxidase activity
A0A4X1W0M9	EF-hand domain-containing protein (S100A16)	0.2025	0.0279	calcium ion binding
A0A4X1SNR4	Ig-like domain-containing protein	0.4225	0.0460	N/A
P12068	Lysozyme C-2	0.4242	0.0075	immunization and immune enhancement
A0A480VWY5	Enoyl-CoA hydratase	0.4587	0.0245	enoyl-CoA hydratase activity
A0A480T775	Bifunctional glutamate/proline-tRNA ligase	0.4717	0.0140	aminoacyl-tRNA synthetase
A0A480XP77	Translation initiation factor eIF-2B subunit delta isoform 2	0.5118	0.0249	translation initiation factor activity
A0A0B8RZL6	Eukaryotic elongation factor 2 kinase (EEF2K)	0.5509	0.0355	calmodulin binding
A0A4X1TGG9	Threonyl-tRNA synthetase (TARS1)	0.5778	0.0404	threonine-tRNA ligase activity
A0A4X1ULA3	HIT domain-containing protein	0.5920	0.0467	catalytic activity

^1^ FC means fold change in ZEN/Control (*n* = 4). N/A indicates that the main function of the protein is not yet clear.

**Table 6 toxins-14-00702-t006:** Proteome Discoverer search parameter.

Items	Value
Proteome Discoverer version:	2.4
Protein Database	uniprot-taxonomy-9823.unique.fasta
Cys alkylation	Iodoacetamide
Dynamic Modification	Oxidation (M), Acetyl (Protein N-Terminus), Met-loss (Protein N-Terminus), Met-loss+ Acetyl (Protein N-Terminus)
Static Modification	Carbamidomethyl (C), TMT 6plex (K), TMT 6plex (N-Terminus)
Enzyme Name	Trypsin (Full)
Max. Missed Cleavage Sites	2
Precursor Mass Tolerance	20 ppm
Fragment Mass Tolerance	0.02 Da
Validation based on	q-value

## Data Availability

In this study, the proteomics data have been deposited to the ProteomeXchange Consortium (http://proteomecentral.proteomexchange.org (accessed on 13 September 2022)) via the iProX partner repository with the dataset identifier PXD036686.

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
