# Peer review of "Quantitative Proteomic Analysis of Zearalenone-Induced Intestinal Damage in Weaned Piglets"

_toxins, 2022, doi:10.3390/toxins14100702_

Round 1

Reviewer 1 Report

The manuscript was well organized, and the topics are well explained,  but small changes are necessary before publication

- Line 365. The sentence “The mycotoxins of the diets were carried out by Qingdao Entry-Exit Inspection and 365 Quarantine Bureau as previously described” is not clear. Please clarify the concept.

- In your materials and methods, there were no Animal Care and Ethics Committee statements, please add.

- Line 371. A full stop is missing.

Author Response

Thank you very much for your thoughtful and useful comments and suggestions. Your comments have improved the manuscript effectively. The parts that reviewers referred have been noted by using red type in the revised manuscript text. We have included almost all of your suggestions and below we present a point-by-point response to your comments.

Point 1: Line 365. The sentence “The mycotoxins of the diets were carried out by Qingdao Entry-Exit Inspection and 365 Quarantine Bureau as previously described” is not clear. Please clarify the concept.

Response 1: Thank you very much for your excellent comment and suggestion. The concept for mycotoxin detection in feed rations for this study has been added in the revised manuscript (Lines 328-333). “The levels of ZEN and aflatoxin were quantified by liquid chromatography in conjunction with fluorescence detection, affinity column chromatography and external standard method, and the contents of fumonisin and deoxynivalenol were quantified by high performance liquid chromatography tandem mass with fluorescence detection, affinity column chromatography and external standard method [14,16].”

Point 2: In your materials and methods, there were no Animal Care and Ethics Committee statements, please add.

Response 2: Thank you very much for your excellent comment and suggestion. The ethical approval for this study has been added in the revised manuscript (Lines 490-493). “Institutional Review Board Statement: This research was conducted at the Animal Research Station of Shandong Agricultural University (Taian, Shandong, China), and the animal protocols for this study was validated by the Institutional Animal Care and Use Committee of Shandong Agricultural University (Approval Number: # SDAUA-2021-0410, Date of approval: 10 April 2021).”

Point 3: Line 371. A full stop is missing.

Response 3: Thank you very much for your excellent comment and suggestion. A full stop“.” was added in the revised manuscript (Line 338).

Reviewer 2 Report

in this paper, the authors exposed Weaned Piglets to 3.0 mg/kg ZEA and explored the Jejunum histology damage as well as change in protein expression and other relevant factors. in my opinion, the experiments are well designed except that the authors used only one dose of ZEA. furthermore, the methods are well explained and written. however, some points need to be clarified further. 

The authors did not mention any ethical approval for this study. how they performed the study without ethical permission?

why authors only applied 3.0 mg/kg ZEA ? only one dose in such studies is not informative enough.

more detail regarding the histological measurement is needed on how authors measured the criteria. 

authors need to discuss how these findings could be environmentally relevant and compare the dosage they used to realistic scenarios.

ZEN exposure in most cases is with DON. authors need to discuss this in the discussion and also discuss the mixture exposure. 

Author Response

Thank you very much for your thoughtful and useful comments and suggestions. Your comments have improved the manuscript effectively. The parts that reviewers referred have been noted by using red type in the revised manuscript text. We have included almost all of your suggestions and below we present a point-by-point response to your comments.

Point 1: The authors did not mention any ethical approval for this study. how they performed the study without ethical permission?

Response 1: Thank you very much for your excellent comment and suggestion. The ethical approval for this study has been added in the revised manuscript (Lines 490-493). “Institutional Review Board Statement: This research was conducted at the Animal Research Station of Shandong Agricultural University (Taian, Shandong, China), and the animal protocols for this study was validated by the Institutional Animal Care and Use Committee of Shandong Agricultural University (Approval Number: # SDAUA-2021-0410, Date of approval: 10 April 2021).”

Point 2: Why authors only applied 3.0 mg/kg ZEA? only one dose in such studies is not informative enough.

Response 2: Thank you very much for your excellent comment and suggestion. Effects of different doses of ZEN in piglet diet on intestinal health have been studies in our previous studies [5,15,16]. Therefore, we aimed to further explore the mechanisms of ZEN-induced intestinal damage using the tandem mass tag proteomics. The dose of ZEN in this study was selected according to the results of our previous study that 3.0 mg/kg of ZEN in piglet diet induced obstruction of intestinal self-repair [16], which has been explained in the revised manuscript (Lines 319-320).

Point 3: More detail regarding the histological measurement is needed on how authors measured the criteria.

Response 3: Thank you very much for your excellent comment and suggestion. The detailed criteria of histological measurement have been added in the revised manuscript. “The detail determination methods of the villi height and the crypt depth were referred to previous study [35]. In short, the villi height was measured from the villi tip to the villi base, and the crypt depth was measured from the intervillous valley to the basement membrane.” (Lines 374-377).

Point 4: Authors need to discuss how these findings could be environmentally relevant and compare the dosage they used to realistic scenarios.

Response 4: Thank you very much for your excellent comment and suggestion. We have discussed this part in the revised manuscript (Lines 202-204). A previous study showed that the positive detection rate of ZEN in feedstuff could reached 69.15%, and the highest value of ZEN in compound feed samples was 4.33 mg/kg [6]. In this study, the dose of ZEN was selected based on our previous studies in which 3.0 mg/kg of ZEN in piglet diet induced obstruction of intestinal self-repair [16]. Our study provides valuable clues to elucidate the possible mechanism of ZEN-induced intestine intestinal injury, and lays the groundwork for future research on ZEN detoxification in animals.

Point 5: ZEN exposure in most cases is with DON. authors need to discuss this in the discussion and also discuss the mixture exposure.

Response 5: Thank you very much for your excellent comment and suggestion. The levels of ZEN and deoxynivalenol (DON) in basal diet were below the minimum detection limits, and the 3.0 mg/kg ZEN in the treatment group was added extra to the diet. Therefore, there was no synergistic effect of ZEN and DON in our experiments.

Reviewer 3 Report

The presented study provides clues for the mechanistic study of ZEA-induced intestinal damage. Although an obvious effort was put in the study, there are some points that should be clarified before it could be submitted:

1-    An ethical approval is not mentioned in the manuscript. An animal study like this cannot be conducted and published without an ethical committee approval. The authors must provide a statement about the ethical approval (e.g. approval number).

2-    The authors selected >1.2 and <0.83 as cutoff values for the fold change (FC). What is the basis for this selection? Setting the FC to classical values (e.g. >2 and <0.5) will dramatically change the output of the study as the number of significantly differentially expressed proteins (DEP) will be dramatically reduced. A brief explanation of how the criteria was set is required.

3-    The authors used the TMT reagent # 90111 from Thermo. This reagent is a 10plex reagent i.e. the set enables up to ten different peptide samples prepared from cells or tissues to be labeled in parallel and then combined for analysis. So why only 8 samples were prepared and analyzed? An additional ZEA and control samples would have improved the statistics of the proteomic data.

4-    In the LC-MS/MS section, the (normalized) collision energy should be mentioned.

5-    A screenshot of the workflow of Proteome Discoverer could be shown in the supporting information.

6-    The English language needs to be checked by a native speaker.

Author Response

Thank you very much for your thoughtful and useful comments and suggestions. Your comments have improved the manuscript effectively. The parts that reviewers referred have been noted by using red type in the revised manuscript text. We have included almost all of your suggestions and below we present a point-by-point response to your comments.

Point 1: An ethical approval is not mentioned in the manuscript. An animal study like this cannot be conducted and published without an ethical committee approval. The authors must provide a statement about the ethical approval (e.g. approval number).

Response 1: Thank you very much for your excellent comment and suggestion. The ethical approval for this study has been added in the revised manuscript (Lines 490-493). “Institutional Review Board Statement: This research was conducted at the Animal Research Station of Shandong Agricultural University (Taian, Shandong, China), and the animal protocols for this study was validated by the Institutional Animal Care and Use Committee of Shandong Agricultural University (Approval Number: # SDAUA-2021-0410, Date of approval: 10 April 2021).”

Point 2: The authors selected >1.2 and <0.83 as cutoff values for the fold change (FC). What is the basis for this selection? Setting the FC to classical values (e.g. >2 and <0.5) will dramatically change the output of the study as the number of significantly differentially expressed proteins (DEP) will be dramatically reduced. A brief explanation of how the criteria was set is required.

Response 2: Thank you very much for your excellent comment and suggestion. We tried several typical FC (e.g. > 2 and < 0.5, > 1.5 and <0.67, > 1.2 and <0.83) and found that FC > 1.2 and < 0.83 was the most appropriate for our data. The FC >1.2 and <0.83 was also generally accepted standard in proteomic analysis (Li et al., 2021; Liu et al., 2022; Ma et al., 2022; Zhang et al., 2021).

Li, X.; Liu, X.; Horvatovich, P.; Hu, Y.; Zhang, J. Proteomics Landscape of Host-Pathogen Interaction in Acinetobacter baumannii Infected Mouse Lung. Front Genet 2021, 12, 563516, DOI:10.3389/fgene.2021.563516.

Liu, X.; An, L.; Ren, S.; Zhou, Y.; Peng, W. Comparative Proteomic Analysis Reveals Antibacterial Mechanism of Patrinia scabiosaefolia Against Methicillin Resistant Staphylococcus epidermidis. Infect Drug Resist 2022, 15, 883-893, DOI:10.2147/IDR.S350715.

Ma, T.L.; Li, W.J.; Hong, Y.S.; Zhou, Y.M.; Tian, L.; Zhang, X.G.; Liu, F.L.; Liu, P. TMT based proteomic profiling of Sophora alopecuroides leaves reveal flavonoid biosynthesis processes in response to salt stress. J Proteomics 2022, 253, 104457, DOI:10.1016/j.jprot.2021.104457.

Zhang, J.; Cao, J.; Geng, A.; Wang, H.; Chu, Q.; Yang, L.; Yan, Z.; Zhang, X.; Zhang, Y.; Dai, J.; et al. Comprehensive Proteomic Characterization of the Pectoralis Major at Three Chronological Ages in Beijing-You Chicken. Front Physiol 2021, 12, 658711, DOI:10.3389/fphys.2021.658711.

Point 3: The authors used the TMT reagent # 90111 from Thermo. This reagent is a 10plex reagent i.e. the set enables up to ten different peptide samples prepared from cells or tissues to be labeled in parallel and then combined for analysis. So why only 8 samples were prepared and analyzed? An additional ZEA and control samples would have improved the statistics of the proteomic data.

Response 3: Thank you very much for your excellent comment and suggestion. Based on preliminary results (e.g. histological and intestinal permeability), four piglets’ intestinal samples from each group were selected for proteomic analysis. Currently, most articles demonstrated that the repetition number of four were sufficiently for analytical requirements in proteomics analysis (Qiu et al., 2018; Li et al., 2021; Liu et al., 2022; Chen et al., 2021; Ma et al., 2022). In our study, we also observed successfully the DEPs that involved in ZEN-induced intestinal injury.

Qiu, K.; Zhang, X.; Wang, L.; Jiao, N.; Xu, D.; Yin, J. Protein Expression Landscape Defines the Differentiation Potential Specificity of Adipogenic and Myogenic Precursors in the Skeletal Muscle. J Proteome Res 2018, 17, 3853-3865, DOI:10.1021/acs.jproteome.8b00530.

Li, X.; Liu, X.; Horvatovich, P.; Hu, Y.; Zhang, J. Proteomics Landscape of Host-Pathogen Interaction in Acinetobacter baumannii Infected Mouse Lung. Front Genet 2021, 12, 563516, DOI:10.3389/fgene.2021.563516.

Liu, X.; An, L.; Ren, S.; Zhou, Y.; Peng, W. Comparative Proteomic Analysis Reveals Antibacterial Mechanism of Patrinia scabiosaefolia Against Methicillin Resistant Staphylococcus epidermidis. Infect Drug Resist 2022, 15, 883-893, DOI:10.2147/IDR.S350715.

Chen, Z.; Zhong, W.; Chen, S.; Zhou, Y.; Ji, P.; Gong, Y.; Yang, Z.; Mao, Z.; Zhang, C.; Mu, F. TMT-based quantitative proteomics analyses of sterile/fertile anthers from a genic male-sterile line and its maintainer in cotton (Gossypium hirsutum L.). J Proteomics 2021, 232, 104026, DOI:10.1016/j.jprot.2020.104026.

Ma, T.L.; Li, W.J.; Hong, Y.S.; Zhou, Y.M.; Tian, L.; Zhang, X.G.; Liu, F.L.; Liu, P. TMT based proteomic profiling of Sophora alopecuroides leaves reveal flavonoid biosynthesis processes in response to salt stress. J Proteomics 2022, 253, 104457, DOI:10.1016/j.jprot.2021.104457.

Point 4: In the LC-MS/MS section, the (normalized) collision energy should be mentioned.

Response 4: Thank you very much for your excellent comment and suggestion. Fragmentation mode was higher-energy collision dissociation with 30% normalized collision energy. The normalized collision energy for the LC-MS/MS has been specified in the revised manuscript (Line 436).

Point 5: A screenshot of the workflow of Proteome Discoverer could be shown in the supporting information.

Response 5: Thank you very much for your excellent comment and suggestion. We have added screenshot of the workflow of Proteome Discoverer to the supporting information. Please see Supplementary Figure S1.

Point 6: The English language needs to be checked by a native speaker.

Response 6: Thank you very much for your excellent comment and suggestion. We have checked our manuscript thoroughly, and corrected some mistakes of language. We sincerely hope that our revised manuscript could meet your requirement.

Reviewer 4 Report

-The actual abbreviation for “zearalenone” is ZEN, not ZEA. I suggest to change ZEA to ZEN

- A more detailed presentation of previous studies on the effects of ZEN on the gastrointestinal tract would increase the value of the article

-Why the authors chose the dose 3 mg/kg. Is it the dose. Whether this is the dose that the human or animal may be exposed to in a daily life? Dose selection should be justified in the materials and methods

-Why was ZEN administered for just 32 days - that should be clarified

-Whether the authors obtained approval from the ethical committee. Please provide the name of the commission, the consent number and the date of its receipt. The lack of consent from the ethics committee prevents publication of the manuscript.

-Are there no more precise methods for determining ET and DAO in serum? Why did the authors choose ELISA tests

-Scale bars on microphotographs are illegible. I suggest to redraft this figure.

-Discussion seems be to long. Some phrases are not connected with results, For example the first paragraph of discussion, which shows general information about ZEN metabolism. I suggest to redraft Discussion.

-What are the limitations of the experiment. Please present them in the discussion.

-The novelty of experiment and eventually practical use of results (for example in animal husbandry) should be included in the conclusions.

-There are some linguistic errors in the text. I suggest checking the entire manuscript carefully.

Author Response

Thank you very much for your thoughtful and useful comments and suggestions. Your comments have improved the manuscript effectively. The parts that reviewers referred have been noted by using red type in the revised manuscript text. We have included almost all of your suggestions and below we present a point-by-point response to your comments.

Point 1: The actual abbreviation for “zearalenone” is ZEN, not ZEA. I suggest to change ZEA to ZEN.

Response 1: Thank you very much for your excellent comment and suggestion. We have changed the “ZEA” to “ZEN” throughout the revised manuscript.

Point 2: A more detailed presentation of previous studies on the effects of ZEN on the gastrointestinal tract would increase the value of the article.

Response 2: Thank you very much for your excellent comment and suggestion. We have added some researches (Ref. 13, 15) on the effects of ZEN on the gastrointestinal tract in the revised manuscript (Lines 37-42). “Saenz et al [13] found that ingestion of ZEN (679 and 1623 μg/kg) could alter gut microbiome by increase intestinal oxidative stress in weaned piglets. Our previous study showed that weaned piglets fed the diets containing 1.04 mg/kg ZEN showed decreased activities of disaccharidase enzymes, reduced the intestinal functional mucosal epithelial surface area, and enhanced oxidative stress in small intestine [14,15].”

Point 3: Why the authors chose the dose 3 mg/kg. Is it the dose. Whether this is the dose that the human or animal may be exposed to in a daily life? Dose selection should be justified in the materials and methods.

Response 3: Thank you very much for your excellent comment and suggestion. Effects of different doses of ZEN in piglet diet on intestinal health have been studies in our previous studies [5,15,16]. Therefore, we aimed to further explore the mechanisms of ZEN-induced intestinal damage using the tandem mass tag proteomics. The dose of ZEN in this study was selected according to the results of our previous study that 3.0 mg/kg of ZEN in piglet diet induced obstruction of intestinal self-repair [16], which has been explained in the revised manuscript (Lines 319-320). A previous study showed that the positive detection rate of ZEN in feedstuff could reached 69.15%, and the highest value of ZEN in compound feed samples was 4.33 mg/kg [6].

Point 4: Why was ZEN administered for just 32 days that should be clarified.

Response 4: Thank you very much for your excellent comment and suggestion. At the late stage of the trial, some piglets in the ZEN group showed dramatic toxic symptom, such as rectal prolapse and severe diarrhea, which made us have to end our trial. Therefore, the trial period lasted 32 days.

Point 5: Whether the authors obtained approval from the ethical committee. Please provide the name of the commission, the consent number and the date of its receipt. The lack of consent from the ethics committee prevents publication of the manuscript.

Response 5: Thank you very much for your excellent comment and suggestion. The ethical approval for this study has been added in the revised manuscript (Lines 490-493). “Institutional Review Board Statement: This research was conducted at the Animal Research Station of Shandong Agricultural University (Taian, Shandong, China), and the animal protocols for this study was validated by the Institutional Animal Care and Use Committee of Shandong Agricultural University (Approval Number: # SDAUA-2021-0410, Date of approval: 10 April 2021).”

Point 6: Are there no more precise methods for determining ET and DAO in serum? Why did the authors choose ELISA test.

Response 6: Thank you very much for your excellent comment and suggestion. The ELISA test is a generally accepted and highly reliable method for the determination of ET and DAO in serum, and has been used in many studies (Chen et al., 2021; Wang et al., 2020; Xue et al., 2018; Cao et al., 2021; Izquierdo-Casas et al., 2018).

Chen, J.L.; Li, F.C.; Yang, W.R.; Jiang, S.Z.; Li, Y. Supplementation with Exogenous Catalase from Penicillium notatum in the Diet Ameliorates Lipopolysaccharide-Induced Intestinal Oxidative Damage through Affecting Intestinal Antioxidant Capacity and Microbiota in Weaned Pigs. Microbiology spectrum 2021 9(3): e00654-21, DOI:10.1128/Spectrum.00654-21.

Wang, Y.; An, Y.; Ma, W.; Yu, H.; Lu, Y.; Zhang, X.; Wang, Y.; Liu, W.; Wang, T.; Xiao, R. 27-Hydroxycholesterol contributes to cognitive deficits in APP/PS1 transgenic mice through microbiota dysbiosis and intestinal barrier dysfunction. J Neuroinflammation 2020, 17, 199, DOI:10.1186/s12974-020-01873-7.

Xue, M.; Ji, X.; Liang, H.; Liu, Y.; Wang, B.; Sun, L.; Li, W. The effect of fucoidan on intestinal flora and intestinal barrier function in rats with breast cancer. Food Funct 2018, 9, 1214-1223, DOI:10.1039/c7fo01677h.

Cao, Y.Y.; Wang, Z.; Wang, Z.H.; Jiang, X.G.; Lu, W.H. Inhibition of miR-155 alleviates sepsis-induced inflammation and intestinal barrier dysfunction by inactivating NF-kappaB signaling. Int Immunopharmacol 2021, 90, 107218, DOI:10.1016/j.intimp.2020.107218.

Izquierdo-Casas, J.; Comas-Baste, O.; Latorre-Moratalla, M.L.; Lorente-Gascon, M.; Duelo, A.; Vidal-Carou, M.C.; Soler-Singla, L. Low serum diamine oxidase (DAO) activity levels in patients with migraine. J Physiol Biochem 2018, 74, 93-99, DOI:10.1007/s13105-017-0571-3.

Point 7: Scale bars on microphotographs are illegible. I suggest to redraft this figure.

Response 7: Thank you very much for your excellent comment and suggestion. We have updated scale bars on microphotographs of Figure 1 in revised manuscript (Line 84).

Point 8: Discussion seems be to long. Some phrases are not connected with results, For example the first paragraph of discussion, which shows general information about ZEN metabolism. I suggest to redraft Discussion.

Response 8: Thank you very much for your excellent comment and suggestion. We have condensed the discussion in the revised manuscript (Lines 200-302).

Point 9: What are the limitations of the experiment. Please present them in the discussion.

Response 9: Thank you very much for your excellent comment and suggestion. Our study showed 3.0 mg/kg of ZEN could lead to the changes of DEPs and intestinal injury in weaned piglets, but precise functions of those DEPs and pathways in ZEN-induced intestinal damage remains to be elucidated by cellular experiments, which has been present in the Discussion part of the revised manuscript (Lines 300-302).

Point 10: The novelty of experiment and eventually practical use of results (for example in animal husbandry) should be included in the conclusions.

Response 10: Thank you very much for your excellent comment and suggestion. The novelty of experiment and eventually practical use of results has been added in the revised manuscript (Lines 309-312). “Although further studies will be required to elucidate the functions of the DEPs, our study provides valuable clues to elucidate the possible mechanism of ZEN-induced intestine intestinal injury, and lays the groundwork for future research on ZEN detoxification in animals.”

Point 11: There are some linguistic errors in the text. I suggest checking the entire manuscript carefully.

Response 11: Thank you very much for your excellent comment and suggestion. We have checked our manuscript thoroughly, and corrected some linguistic errors. We sincerely hope that our revised manuscript could meet your requirement.

Round 2

Reviewer 2 Report

authors provided enough evidence and explanation for my comments. 

Reviewer 4 Report

The Authors took into account all my suggestions.